**Data Availability Statement:** We did not use any data to prepare our manuscript therefore, these

# Establishing a surveillance system on sexual and reproductive health and rights (SRHR) of key populations (KPs) at risk of compromised outcome of SRHR- A protocol for a mixed-method study

**Md. Masud Reza[1], Golam Sarwar[1], Samira Dishti Irfan[1], Mohammad Niaz Morshed Khan[1], A. K. M. Masud Rana[1], Muhammad Manwar Morshed Hemel[1], Mohammad Sha Al Imran[1], Md. Mahbubur Rahman[1], Tanveer Khan Ibne Shafiq[1], Md. Safiullah Sarker[2], Muntasir Alam[2], Mustafizur Rahman[2], Sharful Islam Khan[1] ***

1 Programme for HIV and AIDS, Infectious Diseases Division, International Centre for Diarrhoeal Diseases Research, Bangladesh (icddr,b), Dhaka, Bangladesh, 2 Virology Laboratory, Infectious Diseases Division (IDD), icddr,b, Dhaka, Bangladesh

* sharful@icddrb.org

## Abstract

### Background

Key populations (KPs) who are at risk of compromised situation of sexual and reproductive health and rights in Bangladesh constitute including males having sex with males, male sex workers, transgender women (locally known as hijra) and female sex workers. Globally, these key populations experience various sexual and reproductive health and rights burdens and unmet needs for ailments such as sexually transmitted infections including Neisseria Gonorrhoea, Chlamydia Trachomatis and human papillomavirus. Most key population focused interventions around the world, including Bangladesh, primarily address human immune deficiency virus and sexually transmitted infections-related concerns and provide syndromic management of sexually transmitted infections, other sexual and reproductive health and rights issues are remained overlooked that creates a lack of information in the related areas. There is currently no systematic research in Bangladesh that can produce representative data on sexual and reproductive health and rights among key populations, investigates their sexual and reproductive health and rights needs, how their needs evolve, and investigate underlying factors of sexual and reproductive health and rights issues that is crucial for informing more sexual and reproductive health and rights-friendly interventions for key populations. Keeping all these issues in mind, we are proposing to establish a sexual and reproductive health and rights surveillance system for key populations in Bangladesh.

### Method

The sexual and reproductive health and rights surveillance system will be established in Dhaka for males having sex with males, male sex workers and transgender women, and the

restrictions do not apply to our manuscript. However, the objectives of the proposed SRHR surveillance are to understand the burden, unmet needs and the overall situation of SRHR among MSM, MSW, transgender women (locally known as hijra) and FSW in selected areas in Bangladesh. This type of surveillance study, adopting a mixed-method, is being conducted for the first time not only in Bangladesh but also globally. We will collect data twice in year-1 (2022) and year-2 (2023). In the budget, we kept some money for publications where primary data from this surveillance study will be used to prepare manuscripts. We know that PLOS ONE publishes protocols that reach a variety of readers. Therefore, we thought that we also take an initiative to publish the protocol that may have huge impact in improving the wellbeing and the future design of SRHR services among the targeted population groups in other countries. This is to be noted that as soon as data are ready, we will start work to prepare manuscripts on various SRHR issues of the studied population groups for publications and we hope that we will submit manuscripts to the PLOS ONE.

**Funding:** The authors greatly thank the Department of Foreign Affairs, Trade and Development (DFATD), Canada for the funding of the study (Grant#02063). The authors also thank the Institutional Review Board of icddr,b, for approving the study protocol. We also acknowledge that the funders did not and will not have a role in study design, data collection and analysis, decision to publish, or preparation of the manuscript. This is also to be noted that there is not any separate protocol for the laboratory part which has already been described in the manuscript.

**Competing interests:** The authors have declared that no competing interests exist.

**Abbreviations:** BCC, Behaviour change communication; BFSW, Brothel-based female sex workers; CBO, Community-based organization; CI, Confidence interval; CT, Chlamydia Trachomatis; DIC, Drop-in Centre; FSW, Female sex workers; FRO, Field research officers; FPC, Finite Population Correction; GoB, Government of Bangladesh; HFSW, Hotel-based female sex workers; HIV, Human immunodeficiency virus; HPV, Human papillomavirus; HW, Health workers; ICT, Information communication technology; IDI, In-depth interview; IQR, Interquartile range; KII, Key-informant interview; KP, Key population at risk of HIV; MoHFW, Ministry of Health and Family Welfare; MoU, Memorandum of understanding; MSM, Males having sex with males; MSW, Male sex workers; MT, Medical technologists; NG, Neisseria Gonorrhoea; PCR, Polymerase chain

other in Jashore for female sex workers. The duration will be for 3 years and data will be collected twice, in year one and year two adopting a mixed method repeated cross-sectional design. All key populations 15 years and above will be sampled. Behavioural data will be collected adopting a face-to-face technique and then biological samples will be collected. Those who will be found positive for human papillomavirus, will be referred to a government hospital for treatment. Free treatment will be provided to those who will be found positive for other sexually transmitted infections. In total, 2,240 key populations will be sampled. Written assent/consent will be taken from everyone. Data will be entered by Epi-Info and analysed by Stata. Report will be produced in every year.

## Discussion

This surveillance system will be the first of its kind to systematically assess the situation of sexual and reproductive health and rights among selected key populations in Bangladesh. It is expected that this study will provide insights needed for improving the existing sexual and reproductive health and rights intervention modalities for these vulnerable and marginalized key populations.

## Background

Good sexual and reproductive health (SRH) is defined by the World Health Organisation (WHO) as "a state of complete physical, mental and social well-being in relation to sexuality and reproductive health, and not merely the absence, dysfunction or infirmity" [1]. In addition, United Nations (UN) bodies, e.g. World Health Organizations (WHO), United Nations Population Fund (UNFPA) and United Nations High Commissioner for Refugees (UNHCR) underscore various core tenets of sexual and reproductive health including: positive and respectful sexual relationships, pleasurable and safe sexual experiences, capability and freedom to reproduce, access to safe pregnancy and childbirth services; and access to safe, affordable and acceptable contraceptives [1–3]. To ensure good SRH outcomes irrespective of gender, countries and health systems are responsible for safeguarding people's rights to equitable and accessible information, care and services [1, 2]. Thus, UN bodies and other experts imparted the concept of sexual and reproductive health and rights (SRHR), which has ultimately been integrated into several international covenants and agreements [1, 3].

Sexual and reproductive health and rights (SRHR) is inherently linked with multiple human rights enlisted in the Declaration of Human Rights, such as the right to life and privacy, and freedom from torture, stigma and discrimination [4]. The Special Rapporteur of the United Nations Human Rights Council (UNHRC) outlined the four fundamental principles of SRHR: (a) adequate SRH care services, goods and facilities; (b) physically and economically accessible services; (c) accessible services free of discrimination; (d) services of good quality [3]. Ghebreyesus and colleagues emphasized that the universal access to SRH is crucial for achieving sustainable development and supporting people's human and health rights [5].

Key populations (KPs) at risk of HIV constitute various population groups such as males having sex with males (MSM), male sex workers (MSW), transgender women (locally known as hijra), female sex workers (FSW), and people who inject drugs (PWID; male and female). These KPs experience diverse SRHR burdens such as sexually transmitted infections (STIs) including HIV, Neisseria Gonorrhoea (NG), Chlamydia Trachomatis (CT), human papillomavirus (HPV), cervical cancer, unintended pregnancies, unsafe abortions and gender dysphoria.

reaction; PSU, Primary sampling units; PWID, People who inject drugs; RFSW, Residence-based female sex workers; RPR, Rapid Plasma Reagin; SFSW, Street-based female sex workers; SHG, Self-help group; SRHR, Sexual and reproductive health rights; STI, Sexually transmitted infections; TG, Transgender; TPPA, Treponema palladium particle agglutination; WHO, World Health Organization.

A systematic review found high prevalence rates of anal, penile, and oral HPV infection among MSM (78.4%, 36.2%, and 17.3%, respectively) [6]. A study in Northern Thailand showed that the overall HPV prevalence among transgender women was 80% [7]. A recent systematic review also reported 39.5%-42.6% pooled HPV prevalence among FSW [6, 8]. Another systematic review in lower and middle-income countries revealed a median incidence of 26.8% unintended pregnancies among FSW [9]. The HIV surveillance data in Bangladesh among MSM, MSW, hijra and FSW in 2015–2016 showed active syphilis rates ranging from 1.0%-2.2% [10, 11].

According to the global evidence, these population groups experience SRHR burdens and unmet needs with various implications including mental health concerns, illicit drug use, and limited access to SRHR information and services [12]. This warrants particular attention because of the KPs' heightened exposure to the circumstances which could contribute to SRHR problems such as unprotected sex, multiple sex partners and gender-based violence [13]. As most KP interventions focus on HIV/STI issues, other crucial health domains remain neglected [12]. Moreover, SRHR-related research has mainly focused on women, despite the importance of male sexual health. However, both men and women's needs must be addressed simultaneously to enhance their quality of life [14]. Therefore, it is integral that SRHR-related concerns are adequately researched and intervened, especially among those who experience healthcare access barriers.

Despite being an overall low HIV-prevalent country for general populations, the prevalence was 1.5% for KPs as of 2020 in the districts containing HIV prevention services [15], which constituted one third of the HIV burden [16]. There is generally a scarcity of information about the SRHR situation among the KPs in Bangladesh, especially among MSM and hijra because of their hidden and marginalised nature. A cross-sectional survey by Wahed and colleagues in 2017 demonstrated that the majority of the FSW (84.4%) had a childbirth experience at least once in their lifetime and 52% of them had an abortion [17]. Similarly, an article on FSW by Katz and colleagues (2015) revealed that 44% of the hotel-based FSW and 30% of the street-based FSW in Dhaka, Bangladesh resorted to dual contraceptive methods (consistent condom use with a non-barrier contraceptive) [18].

Thus, establishing a surveillance system of SRHR among KPs could engender several benefits which would ultimately guide designing SRHR services in Bangladesh which could be integrated not only within the existing KP intervention modalities but also within the healthcare settings of the public health care facilities.

## Objectives

**Primary objective.** To establish a surveillance system for SRHR among selected KPs in Bangladesh.

**Secondary objectives.**

1. To estimate the prevalence of selected SRHR indicators (biological, behavioral and socio-demographics, provided in Annex 2 in S1 File).

2. To identify emerging SRHR health problems among KPs to generate priority areas for research and interventions.

3. To assess study participants' knowledge on transmission and prevention of COVID-19 and impact of COVID-19 in the utilization of SRHR services.

4. To generate evidence-based recommendations and conduct policy dialogue for interventions on SRHR with KPs.

## Methods

As this surveillance system for KPs will not only measure the prevalence of selected SRHR indicators but also will produce evidence of changes of important indicators between two time points. The methodologies of the proposed surveillance system for KPs are described in detail below.

### Research design

The proposed SRHR surveillance will follow a repeated cross-sectional survey methodology that will be blended with qualitative methods. The quantitative method will be used to measure the prevalence of selected SRHR indicators in every year and then to measure the changes in the selected SRHR indicators between two time points, baseline and the subsequent round. During quantitative data collection, qualitative methods will also be used to understand and explain quantitative data. After the end of two rounds of quantitative data collection, qualitative methods will also be used to explore the reasons for changes in the quantitative parameters.

### Definition of KPs

In the SRHR surveillance system of KPs, we are proposing four groups to be included. These are MSM, MSW, hijra and FSW (street based, hotel based, residence based and brothel based). Since we are focusing on SRHR which are associated with sexual health, sexuality and sexual behaviours, and reproductive health related issues of KPs therefore, we have decided to exclude people who inject drugs (PWID, male and female) as their risks are more drug use-oriented. The definition of KPs in the surveillance system is defined as per the national document as follows [10, 11], Table 1.

### Inclusion criteria

MSM, MSW, hijra and FSW will be included in the SRHR surveillance if they meet the definitions shown in Table 1 above and if they are:

- Age 15 years or older, and provide

- Written informed assent/consent

**Table 1. Definition of KPs.**

| KPs | Definition |
| --- | --- |
| Males who have sex with males (MSM) | Males who have sex with males but did not sell sex in the last one year |
| Male Sex Workers (MSW) | Males who sell sex in exchange of money or compulsory gift in the last one month |
| Transgender women (hijra) | Who identify themselves as belonging to a traditional hijra sub-culture |
| Female sex workers (FSW) | |
| Street based FSW (SFSW) | Those who were contracted by clients on the street, with the sex act taking place in a public space or other venues in the last one month |
| Hotel based FSW (HFSW) | Those who were contracted by clients in a hotel setting, with the sex act taking place in hotels in the last one month |
| Residence based FSW (RFSW) | Those who identified themselves as sex workers and sold sex in residences in the last one month |
| Brothel based FSW (BFSW) | Those who were contracted by clients in a brothel setting, with the sex act generally taking place in brothels in the last one month |

○ Assent will be taken from those who will be 15 to less than 18 years of age

○ Consent will be taken from those who will be 18 or above years of age

## Duration of SRHR surveillance system and sampling sites

The surveillance system will span three years (April 2022-March 2025) and target MSM, MSW and hijra in Dhaka, as well as FSW (i.e. street, residence, hotel and brothel) in Jashore (a district in the western part of Bangladesh close to Indian boarder). Sample sizes were determined by comparing estimated KP sizes in Dhaka, Chittagong, Sylhet, Khulna and Jashore districts with the required sample size [19] (Annex 1 in S1 File). Thereafter, Dhaka and Jashore was selected as SRHR surveillance sites not only due to availability of KPs abundantly but also spots (defined in Table 3) are geographically well defined, existence of NGOs/CBOs to establish network with the KPs to ensure their participation to provide data face to face and communicate with them to provide free treatment for STIs. However, this is to be acknowledged the data that would be generated from these two sites may not be representative of all MSM, MSW, hijra and FSW in Bangladesh.

## SRHR indicators to be measured among KPs

SRHR indicators will be measured at the individual, facility and national/policy levels. At the initial stage of selecting individual SRHR indicators for KPs, we reviewed some global and national literature such as WHO's short list of reproductive health indicators for global monitoring [20] and on universal access to sexual and reproductive rights: Bangladesh [21]. After gaining insights from these documents, we have also added some corona virus disease of 2019 (COVID-19) related indicators. In total, we have selected thirty-nine indicators (i.e., 28 individual, eight facility, and three at policy levels). To avoid overlap with the existing HIV prevention programme for KPs funded by the Government of Bangladesh and the Global Fund, we did not include one SRHR indicator, i.e., the prevalence of HIV among KPs. The prevalence of HIV is measured at the HIV surveillance rounds conducted by AIDS/STD Programme (ASP), Government of Bangladesh every two years. The list of SRHR indicators to be measured among KPs at the individual, facility, national/policy level is provided in Annex 2 in S1 File.

## Frequency of data collection

All indicators will be measured once in every year for the first two years to establish the surveillance system, and then in the second year, along with the qualitative exploration, we will recommend evidence-based policy with SRHR for the KPs, translate knowledge into actions and help design SRHR interventions for these marginalised population groups.

**Sample size calculation and outcome variables.** Based on the indicators to assess SRHR situation among KPs mentioned in Annex 2 in S1 File, the outcome variables of the proposed SRHR surveillance are:

## Primary (to be measured at the individual level)

1. Prevalence of active syphilis

2. Prevalence of HPV (high risk and low risk subtypes)

3. Prevalence of NG and CT

4. Knowledge of SRHR and HIV

5. Currently use any modern contraceptive method to limit child bearing

6. Used condom during last sex act with non-transactional and transactional male/hijra/ female sex partners to prevent HIV and STIs

7. Decision making on condom during last sex act (alone or jointly)

8. Decision making on family planning methods (alone or jointly)

9. Received services to manage adverse effects of using steroids/hormones

10. FSWs attended at least four antenatal care visits during last pregnancy

11. Births in FSW attended by skilled birth attendant

12. Births in FSW that are reported as unintended

13. Knowledge on Covid-19 transmission and prevention

### Secondary (to be measured at the facility level)

1. Utilisation of essential SRHR, maternal and new-born services in public/private/NGO health facilities

2. Number of health care service providers trained in essential SRHR services

In order to measure the above-mentioned individual level outcome variables, the calculations of sample sizes are described below.

### Sample size calculations for the outcome variables to be measured at the individual level

Once the outcome variables were selected from the list of the SRHR indicators for the KPs (Annex 2 in S1 File), we searched for data values from published literature and reports. However, most indicators had no data available in Bangladesh or other countries. Sample sizes were calculated properly for each of the indicators and KPs (Annex 3 in S1 File) using a standard formula mentioned below [22] with 0.5–5% precision, 95% confidence interval (CI) and design effect of 1.50. Sample sizes were adjusted for Finite Population Correction (FPC) [23] and then inflated by 5% to adjust for refusal/non-response. This is to be noted while applying FPC in formula-2, the data on the size of the KPs were extracted from the report of national size estimation of KPs in Bangladesh [19]. Finally, the highest value from the calculated sample sizes within each KP was chosen to conduct the baseline survey. The maximum sample size for FSW 513, for MSM 566, for MSW 551 and for hijra 610. In total, 2,240 KPs will be sampled at the baseline from two study sites, Dhaka and Jashore.

This is to be mentioned that the calculated sample sizes were based on few indicators and will be applied only to conduct the baseline survey in the study sites for each of the KPs. After having data from the baseline survey on all SRHR individual level indicators for KPs (Annex 2 in S1 File), in order to conduct the last round of survey, data will be taken from the baseline survey to re-calculate sample sizes using a standard formula [24]. This is also to be mentioned that utmost care will be taken in choosing the values of the parameter values in re-calculating sample sizes using formula-3 so that the sample sizes in the second round are similar to the baseline as much as possible in order to ensure minimum impact on the budget line.

$$n = D \frac{z_{1-\alpha/2}^2}{d^2} pq$$

In the above equation:

n = Calculated sample size

D = Design effect

p = Estimated percentage points of the indicators at the time from which the data were extracted to conduct baseline survey

q = 1-p

$Z_{\alpha/2}$ = Two tailed Z-score corresponding to the desired level of significance = 1.96 (with 95% CI)

d = Desired level of precision

## Sample size calculations for the outcome variables to be measured at the facility level

In order to measure utilisation of essential SRHR, maternal and new-born services in public/private/NGO health facilities by the KPs, first of all, a list of health facilities (public/private/NGO) will be prepared both in Dhaka and Jashore. From the list, one third of the facilities from each of the modalities (public/private/NGO) will be selected randomly to visit to collect required information.

## Sampling methods and procedures (quantitative)

Based on the availability of KPs where they congregate to sell or buy sex or to meet each other, two types of sampling techniques will be adopted to conduct face-to-face interview with the study participants both at the baseline and in the last round that is described below.

### a) Two-stage cluster sampling method for MSM/MSW/FSW (street/hotel/residence based):

For data collection from KPs (baseline and the second round), Time Location Sampling (TLS) will be used, a two-stage cluster sampling technique [10, 11, 24]. All KPs will be interviewed at specific spots (defined in Table 2). In the first stage of sampling, a social mapping exercise will be conducted to identify 'spots' or Primary Sampling Units (PSUs) where the members of the particular KPs are available in a particular time frame in a day, such as 7pm to 10pm. The timing of the spots will be determined by visiting them and gathering information from secondary sources like gatekeepers, service providers, and prior HIV surveillance data. This mapping exercise will generate a list of spots in a specific geographic area, including details on their timing, number of KPs observed, gatekeepers' names, and peak availability of KPs. The collected mapping data will be entered into Excel and checked for inconsistencies before preparing a clean version for the next sampling step.

In the second stage of sampling, in order to meet the target sample size in each KP, spots will be chosen adopting a systematic random sampling technique. Thereafter, the spots will be visited and a fixed number of KPs will be interviewed until the target sample size is met [10,

**Table 2. Definition of the spot/PSU from where KPs will be sampled [10, 11].**

| Key Populations | Definition of a spot/PSU |
|---|---|
| Street FSW | A specific location where at least 3 FSW are found in a specific time frame who sell sex |
| Hotel FSW | A residential hotel where at least 5 FSW are found in a specific time frame who sell sex |
| Residence FSW | A specific house where at least 5 FSW are found in a specific time frame who sell sex |
| Brothel FSW | A specific room used in a brothel for selling sex |
| MSM | A specific location where at least 3 MSM are found during a specific time frame |
| MSW | A specific location where at least 3 MSW are found during a specific time frame who sell sex |
| Hijra | A house where at least 3 hijra are living |

11, 24]. This is important to be mentioned that since the number of KPs in the spots vary every day and week therefore, probability proportional to size (PPS) sampling technique will not be feasible and hence a fixed number of KPs will be interviewed [24]. If the number of KPs seen during mapping is less than the calculated sample size, a 'take-all' sampling approach will be adopted otherwise a fixed number of KPs will be selected for interview as mentioned above [10, 11, 24].

In conducting mapping, utmost care will be taken to cover those spots where the members of MSM/MSW/FSW (street/hotel/residence based) gather abundantly. The survey team will collect all information with the help of local guides/peers and the information will be recorded on a hard copy. The members of the survey team will also apply their own judgment to explore new spots and will take notes if any spot is closed or shifted to a new place so that it can also be visited. The study team members will also take measures to control duplications of KPs between or among spots as was applied in conducting HIV surveillance rounds in the past years [10, 11, 25–27].

**b) Systematic proportionate random sampling for brothel based FSW and hijra:**

## For brothel based FSW

There are two brothels in Jashore. In a brothel, the PSU is defined as a room with active FSW living in that room therefore, the total number of used rooms in each brothel by the active FSW will be counted and then the number of active FSW in each room will also be counted. From this mapping exercise, a list of all active FSW in brothels will be obtained and a proportionate allocation of the target sample size will be applied to each brothel [11, 24]. All brothels in the study area will be visited for mapping and survey data collection.

## For hijra

The social mapping method used will conduct a census of hijra in the study area. This method, known as the "birit-based method" was approved by the Government of Bangladesh [28] for HIV surveillance rounds in Dhaka [10, 15]. Hijra Gurus operate within specific catchment areas called "birits" where all their chela activities take place. In the first stage of sampling, a mapping exercise will be carried out during a meeting with the hijra community members to create a list of Gurus. Subsequently, the team will visit each Guru's residence to gather information on the number of chela under their birit. This information will be recorded in a prescribed format, serving as the sampling frame for hijra Gurus and the size of chela in Dhaka city. In the second stage of sampling, the target sample size will be distributed proportionately among the catchment areas (birits) of all Gurus for interviews and collection of biological samples.

This is to be noted that in each round of data collection, a separate mapping exercise will be conducted for each type of KPs and then data will be collected adopting similar sampling techniques as mentioned above. In each round, separate mapping of FSWs will be conducted on the street, hotel, residence and brothels and then the target sample size will be proportionately sub-divided among all types of FSWs based on the number counted during mapping. To conduct mapping and survey data collection, staff members from the community of MSM/MSW/hijra/FSW will be recruited on the basis of prior experience in conducting HIV surveillance rounds in the past in Bangladesh.

## Sampling methods and procedures (qualitative)

Qualitative methodologies will be employed to identify emerging SRHR problems among KPs, interpret and explain quantitative data and formulate evidence-based recommendations for

policy dialogue on SRHR interventions for KPs. In-depth interviews (IDIs) will be conducted using the maximum variation sampling approach to obtain a comprehensive understanding of participants and cross-cutting themes across different socio-demographic and KP groups [29]. This approach helps identify commonalities and discrepancies among participants based on age, marital status, living status, occupation, etc. An estimated total of 20–25 IDIs will be undertaken with different groups of KPs (i.e., FSW, MSM, MSW and hijra). The numbers may vary based on the points of data redundancy and saturation, as well as emerging issues. A semi-structured interview guideline will be used, allowing flexibility to adapt to changes in the field situation.

To gather recommendations for policy dialogue, key-informant interviews (KIIs) will be conducted following an intensity of information approach. Key-informants, including experienced managers, policy planners, government representatives (DGHS), UN bodies, CBOs, NGO leaders, SRHR research experts, human rights activists, and legal representatives, will interpret quantitative data and propose evidence-based recommendations. Potential informants will be identified, interviews scheduled, and 10–15 KIIs conducted. The number may vary based on field changes. KIIs will follow specific guidelines.

## Place of behavioural data collection

All interviews will be held at the spots for MSM/MSW/street FSW, for hijra, at their residence where they live with other hijra population which is separated from their family members, for hotel FSW, at the hotels, for residence FSW, at the residence where they sell sex and for brothels, at the brothels. All interviews will be held maintaining confidentiality and privacy.

## Laboratory methods

From four study population (MSM, MSW, hijra and FSW) different specimens will be collected which is relevant to their sexual practice. Target organisms will be *Neisseria gonorrhoea (NG)*, *Chlamydia trachomatis (CT)*, and human papillomavirus (HPV). In addition, syphilis will be tested using blood sample (Table 3 and Fig 1).

## Collection of specimens

MSW and hijra population will be approached for anorectal and oropharyngeal swab samples. These swabs will be collected in commercial M4RT transport medium. From MSM population urine and from FSW population, cervical swab specimens will be collected, respectively. Trained medical technologist /physician will collect the specimens following aseptic techniques. The specimens in M4RT transport medium will be used to detect CT, NG and HPV

**Table 3. Sampling population, site and test.**

| Population | Sample type | | | | | Test | | | | |
|---|---|---|---|---|---|---|---|---|---|---|
| | | | | | | PCR | | | Culture | Syphilis |
| | **Anorectal swab** | **Oro-pharyngeal swab** | **Urine** | **Cervical swab** | **Blood** | **CT** | **HPV** | **NG** | **NG** | **RPR -1** |
| MSW | 551 | 551* | - | - | 551 | 1102 | 551 | 1102 | 100 | 551 |
| Hijra | 610 | 610* | - | - | 610 | 1220 | 610 | 1220 | 100 | 610 |
| MSM | - | - | 566 | - | 566 | 566 | 566 | 566 | - | 566 |
| FSW | - | - | - | 513 | 513 | 513 | 513 | 513 | 275 | 513 |
| **Total** | **1,161** | **1,161** | **566** | **513** | **2,240** | **3,401** | **2,240** | **3,401** | **475** | **2,240** |

* Oro-pharyngeal swab will be used for detection of CT/NG only not for HPV testing

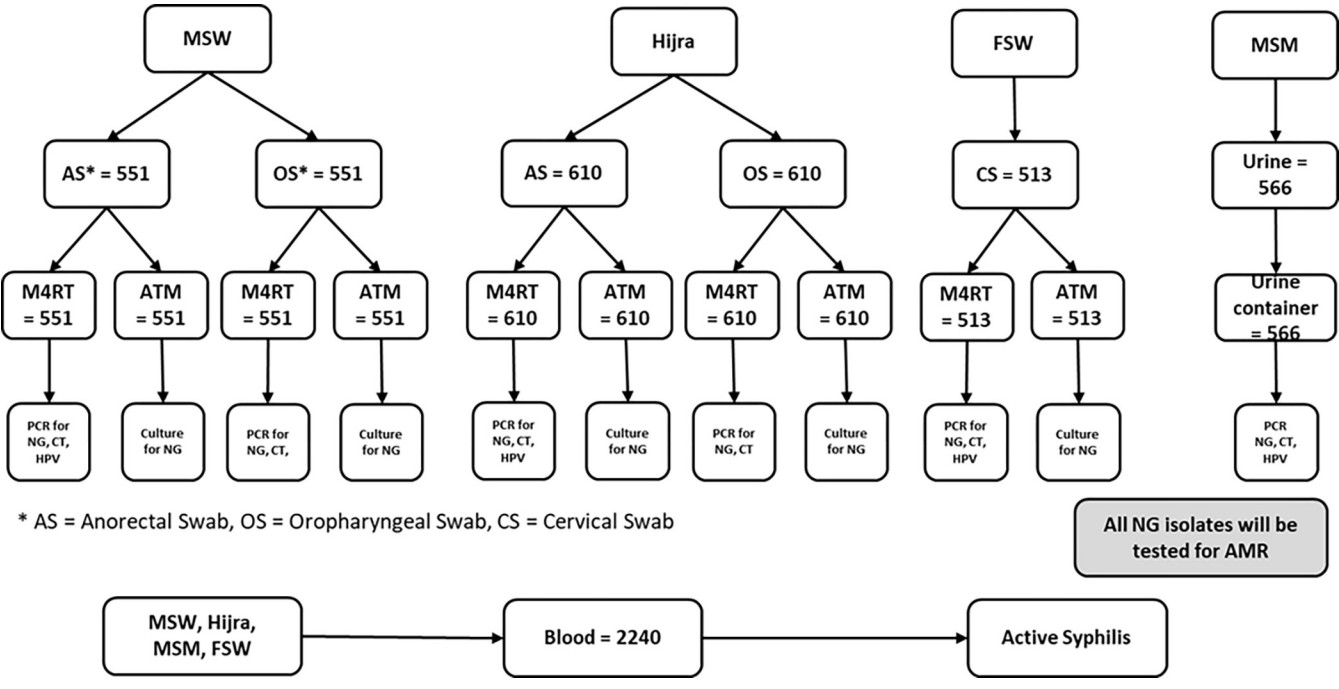

**Fig 1. Specimen collection and laboratory diagnosis outline.**

using molecular techniques (Polymerase Chain Reaction). For N. gonorrhoea culture, an additional cervical swab from all FSW participants, and anorectal and oropharyngeal swab from all MSW and hijra participants will be collected in commercial Amies transport medium containing charcoal. Urine samples from MSM will be collected in a sterile plastic container. For detection of syphilis, 3 mL blood will be collected from all participants by venepuncture into sterile, plain vacutainers (Becton Dickinson, Rutherford, NJ). Serum will be separated from blood using standard protocol on site. By maintaining the cold chain, the laboratory specimens will be transported to the Virology Laboratory of icddr,b.

## Identification of target microbial population

**N. gonorrhoea detection by bacterial culture.** Microbial culture for *N. gonorrhoea* will be performed by inoculating the collected swab in Amies transport medium onto modified Thayer Martin agar media and incubated under microaerophilic conditions (5% $O_2$) for a minimum of 48 hours at a temperature of 35˚C. After incubation, suspected colonies will be sent to Clinical Microbiology Laboratory of icddr,b for confirmed identification and antimicrobial susceptibility tests for *N. gonorrhoea* following CLSI guidelines [30].

**Detection of targeted pathogens using molecular methods.** Samples collected in M4RT transport media will be used for identifying HPV, *N. gonorrhoea* and *C. trachomatis* through the polymerase chain reaction from anorectal swab, urine, oropharyngeal, and cervical swab. This is to be noted that anorectal, urine and cervical samples will be collected in the first and second years. DNA will be first extracted using commercial Qiagen nucleic acid extraction kit (Qiagen GmbH, Germany), and PCR will be performed by using specific primers targeting *C. trachomatis Orf8* gene and *N. gonorrhoea 16S rRNA* gene (PMID: 27898572). HPV genotyping will be carried out using 23 HPV genotyping Real-time PCR kit (HBRT-23, Hybribio HPV Detection Kit, Hybribio Ltd, Wanchai, Hong Kong). This kit can detect 15 high-risk types

(type 31, 33, 35, 39, 45, 51, 52, 53, 56, 58, 59, 66, 68, 73, 82), 2 common high-risk types (type 16, 18) and 6 low-risk types of HPV (type 6, 11, 42, 43, 44, 81).

**Human Papillomavirus (HPV) detection from urine samples.** A 6ml aliquot urine sample will be centrifuged at 5000 rpm for a period of 15 minutes. The pellet will be resuspended in a 500 µl phosphate buffer saline upon decantation of the supernatant. DNA extraction will be conducted using a 200 µl aliquot of this concentrated sample using a DNeasy Blood and Tissue Kit (Qiagen, Hilden, Germany) as per the instructions of the manufacturer. The extracted DNA will be stored at −20˚C until further testing. HPV genotyping will be carried out using similar method described for a swab.

**Syphilis detection.** Syphilis will be detected from processed serum samples using Treponema pallidum Particle Agglutination (TPPA) test (Serodia TPPA, Fujirebio Inc., Japan) and the Rapid Plasma Reagin (RPR) test (Nostion II, Biomerieux BV, Boxtel, The Netherlands). The samples that are tested positive for both TPPA and RPR for ever having syphilis and those with an RPR titre of ≥8 will be considered active syphilis.

## Treatment arrangement and referral mechanism

The participants who tested positive for any of the STIs (HPV/NG/CT/Active syphilis) will receive free treatment (either directly or through appropriate referral) from the project. The culture test result for N. gonorrhoea will be available within 3 to 4 days of sample collection. PCR for testing of gonorrhoea, chlamydia and HPV will take at least 5 days to complete the test and result to be available. If laboratory result is positive for gonorrhoea, chlamydia and active syphilis, the project physician will provide free treatment from the project following the national STI management guideline. In case of HPV positive FSW, she will be referred to gynae-oncology department of Jashore Medical College and Hospital (JMCH) for further management. In case of HPV positive of MSM, MSW and hijra, referral will be done to Skin & VD department (outdoor) of Bangabandhu Sheik Mujib Medical College and University (BSMMU) in Dhaka. All referrals will be accompanied through a project staff. A functional referral system will be developed with relevant facilities prior to the project implementation.

## Training

A training programme for approximately for 3 weeks (3x5 = 15 working days) will be organised for the recruited staff members at icddr,b. The training will be conducted according to the following manners:

Staff members, including research investigators, field research officers (FRO), data collectors, physicians, counsellors, and medical technologists (MT), will undergo comprehensive training. This training will cover reproductive health, STIs, sexual dynamics of KPs and their partners, sensitivity to the population groups involved, available reproductive health services, and ethical considerations. They will also receive training on interviewing techniques using a semi-structured questionnaire. Three physicians (two males and one female) will be trained to collect biological samples from MSM, MSW, hijra, and FSW. The male physicians will collect urine, ano-rectal, and oropharyngeal samples, while the female physician will collect cervical samples. The physicians' training will be provided by experienced physicians from the Programme for HIV, icddr,b, including practical sessions for sample collection. Three MTs will receive training on blood collection for syphilis testing, serum separation, and sample preservation. This training will be supervised by the virology laboratory, icddr,b. The MTs will also be trained on transporting samples from the field office to the virology laboratory. Confidentiality, respect for participants, biosafety, and biosecurity will be emphasized throughout the training by relevant experts from the Programme for HIV.

At the end of each of the training sessions, field testing of obtaining assent/consent, face to face interview, collection of biological samples will be conducted. Field testing of data collection from FSW will be conducted in Dhaka, from MSM/MSW in Gazipur and from hijra in Dhamrai. During field testing, existing drop-in-centres (DIC) that are being run by icddr,b funded by the Global Fund will be used. Experience from the field testing will be used to fine-tune quality of work in collecting biological samples, and face to face interview that will enhance validity of data and smooth running of the study.

## Innovation in the proposed SRHR surveillance system for KPs: Integration of qualitative and quantitative approaches

Qualitative approaches are seldom used in surveillance systems which predominantly focus on quantitative indicators for disease patterns and magnitudes. However, incorporating qualitative approaches can provide valuable insights and enhance evidence-based programming. To achieve this, we will hold a meeting with experts, NGOs, and CBOs experienced in providing SRHR services to KPs. The meeting will discuss indicator selection, precise data collection during mapping and surveys, and appropriate compensation for participants. Qualitative methods will be employed during quantitative data collection to understand and explain the obtained results. This initiative will enable us to effectively interpret and refine the indicators, ensuring contextualized data interpretation. A group of qualitative researchers will train staff members in collecting qualitative data. This mixed-methods approach transcends the conventional surveillance framework and is depicted in Fig 2.

## Evidence to actions: Modified framework of knowledge translation

Public health experts highlighted that knowledge derived from research and experience lacks value unless transposed into actionable evidence [31]. In this context, the notion of knowledge translation has emerged as a paradigm to address public health challenges and supplement knowledge gaps [31]. Defined as "the synthesis, exchange and application of knowledge by

**Fig 2. Blending of methodologies: Surveillance on SRHR for KPs.**

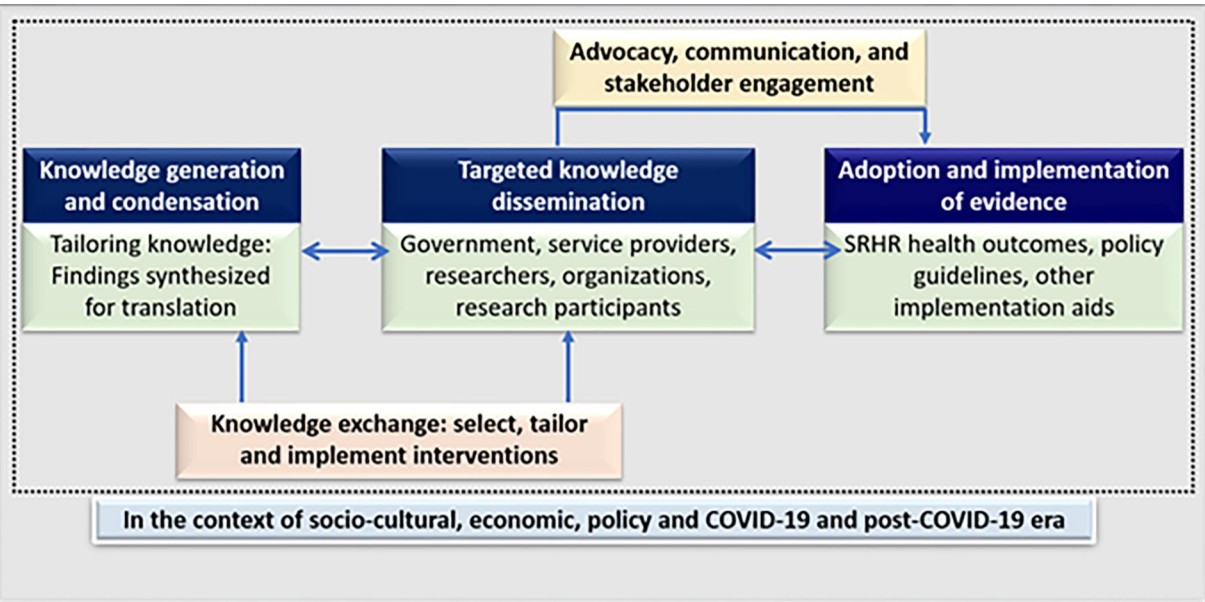

**Fig 3. Evidence to action through a modified knowledge translation framework.**

relevant stakeholders" knowledge translation is posited to expedite the benefits of research innovations for improving health outcomes and health systems [31]. In this context, to construct the foundation for contextualised and effective SRH interventions for KPs, we plan to adopt the knowledge translation framework for the surveillance findings. In this process (illustrated in Fig 3), the knowledge generated from the surveillance will be synthesised and condensed into a lay summary which can be easily exchanged and articulated in a non-scientific setting. Thereafter, targeted knowledge dissemination initiatives will be conducted with selected stakeholders such as government stakeholders, service providers, researchers, organizations and study community members. Through continuous advocacy, communication and stakeholder engagement, evidence will be put into practice through two main forms: (a) incorporation into the existing KP interventions and healthcare setups; and (b) future SRH interventions and research activities taken up by donors or governmental bodies. It is expected that this paradigm will improve SRH outcomes, facilitate its incorporation into policy guidelines and strengthen the overall SRH landscape for vulnerable, marginalised populations like KPs.

### Risk behaviour questionnaires for behavioural surveillance

All risk behavioural data from the KPs will be collected by adopting face to face interview using a semi-structured questionnaire in Bengali.

### Timeline

The work plan based on the above activities with the time frame is shown in Fig 4.

### Data analysis

**Quantitative.** The result of biological samples will be released from the virology laboratory for each of the members in each KP with a unique ID so that this can be matched and merged with the data file of socio-demographics and other variables related to SRHR before data are analysed. For each KP, biological data will be prepared by Excel and data on socio-

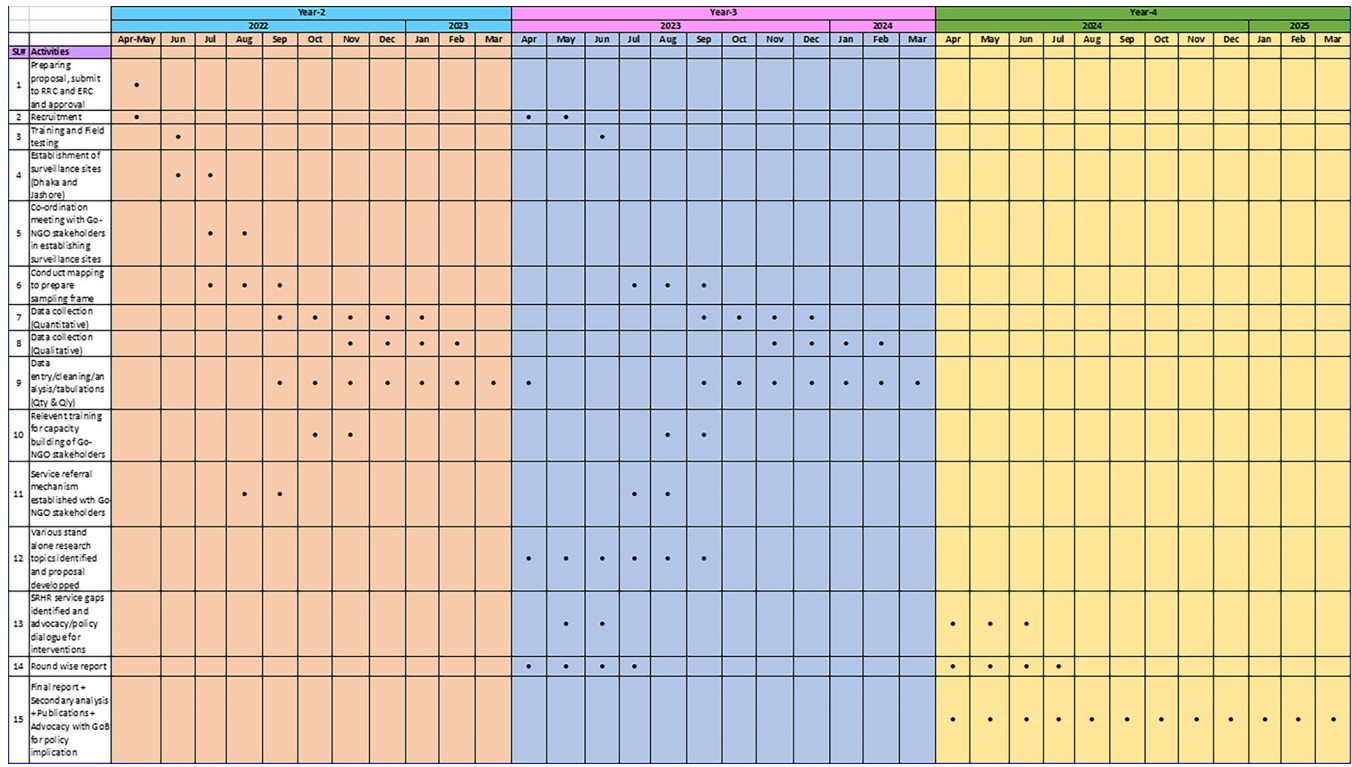

**Fig 4. Workplan.**

demographics and other variables related to SRHR will be entered using Epi-Info for Windows (Version 7.2.4.0); range and consistency checks will be incorporated in the data entry screens. Thereafter, all data files will be converted to Excel for further cleaning by filtering to check consistencies of denominators and responses to the questions asked. All categorical variables will be expressed in terms of percentage points and numerical variables by mean with standard deviation (if normally distributed) or by median with inter-quartile range (IQR) (if not normally distributed).

Clustering of observations will be incorporated in the calculation of 95% confidence interval (CI) for mean and percentage points [24]. During survey, the number of KPs seen during interview in each of the spots will be collected to be used in the calculation of sampling weights to adjust selection biases [24]. In case of a 'Take-all' sampling technique, sampling weights will be 1.0 [24]. In each KP, all statistics will be produced in-terms of age groups 15–24 and 25+ to find out the inequality of the status of the SRHR indicators mentioned in Annex 2 in S1 File. In each KP, all indicators will also be compared between baseline and the second round using Chi-square statistics for categorial variables and using t-test for numeric variables (if normally distributed) or using non-parametric statistics (if not normally distributed). Data will be analysed using Stata (Version 15.1). Facility and national/policy level indicators will also be measured once a year and will be expressed in-terms of percentage points or number. The relationship of the objectives, outcome variables and the method of measurement is provided in Annex 4 in S1 File.

**Qualitative.** The analysis of the qualitative data emerging from the IDIs and KIIs will be converged with the quantitative analysis, thus fostering a deeper understanding of the SRH needs and complexities among KPs. The IDIs and KIIs from the surveillance study will be

digitally recorded upon receiving the written consents of the participants. Data will be systematically stored on a password-protected computer, as per the requirements of the Institutional Review Board and data policy of icddr,b. As qualitative data collection and analysis are ongoing and reflexive processes [32], these two will be conducted concurrently. This would allow for the modification of the interview guidelines, where necessary, and help identify data redundancies and saturation.

Field team members will transcribe recorded interviews on the same day of data collection. Although the interviews will be conducted in Bengali, it is assumed that the data will also contain local dialects and terminologies from the KP communities. Therefore, the field researchers will carefully listen to the recorded data, explore and clarify the meanings of local phrases. When the findings are being reported, the local terminologies will be retained to reflect the cultural essence of the study community. Each interview will be assigned a unique identifying number.

The research team will pursue the six steps of thematic analysis stipulated by Braun and Clarke [33]. The field team members will repeatedly peruse a small subset of the interview transcripts, and try to identify the themes and sub-themes. These themes will formulate the basis of a thematic matrix which will initially be applied to the remainder of the data. The contexts and meanings of these themes and sub-themes will be further analysed. Qualitative data will be sorted by reading the transcripts, and emerging and re-emerging issues which have not been considered beforehand. These issues will be subsequently incorporated into the interview guidelines and data gaps will be supplemented through ongoing field visits. Field researchers will also maintain field diaries in which they will document important observations [33]. The field notes will be analysed in a similar manner to the interview transcripts. Atypical or diverse data will not be ignored. Rather, they will be further explored and presented as findings.

To ensure the scientific rigor of the qualitative research, several approaches will be used, keeping in mind the principles of dependability, confirmability, transferability and credibility stipulated by Lincoln and Guba [34]. These four pillars constitute the qualitative analogue of validity and reliability in qualitative research. To achieve this, we will adopt triangulation, i.e., the use of multiple data collection sources, techniques, investigators and analytical approaches [29]. In addition, peer debriefing sessions will be held to exchange and deliberate about findings and interpretations [35]. Moreover, member-checking sessions will be held to ensure the correct interpretation of the findings from the emic perspectives of the KPs [35].

## Ethical assurance for protection of human rights

Informed written assent will be taken from the participants of 15 to less than 18 years of age and informed written consent will be taken from those who will be 18 years or above years of age. Written assent will be taken from local gatekeepers (such as, hawkers in the spots or someone who belongs to the community of the participant and at the same time knew the participant very well and older in age, at least 18 years, than the participant) whose help will be sought to identify participants. The assent will be read out to the participant and gatekeeper and their signature or left thumb print impression will be taken on the survey form. This is to be mentioned that in the context of Bangladesh, the target population groups are highly stigmatised and their sexual behaviour (male-to-male sex) is not only culturally unaccepted but also prohibited by law so that there is no way we can identify and conduct face-to-face interview at their house where they live with their parents/siblings/other family members. Therefore, based on the context of Bangladesh, the process of taking consent and assent that is being followed in our study is being practised in conducing HIV surveillance rounds among KPs in Bangladesh over the years in the past [10, 11, 15, 25–27].

A unique identification number (ID) will be given to each study participant which will not be linked to the name of the individual. The PSU/DIC that will be used for sample collection will also be coded. All questionnaires will be delinked and will contain the code for PSU/DIC and unique ID of the study participants. In the biological sample collection register, similar unique IDs will be used. In addition, name, address, mobile number and mother list id (if he/ she is enlisted in any DIC to receive HIV prevention services) will be collected in order to contact the participant to provide free treatment for active syphilis/NG/CT and referral for HPV. Utmost care will be taken to maintain confidentiality of the information collected. All sample containers will also be uniquely coded as of interview questionnaire.

All filled-out questionnaires will be kept in locked cabinets in the icddr,b Project Office in Dhaka. The cabinets will only be accessible to the investigators of the study. All computers containing data will be password protected. All interviews and sample collection will be conducted in a private space where the participant is comfortable.

Biological samples will be collected between 6pm-9pm as soon as behavioural interview is completed outside the DIC/at a designated place and hence, conveyance (transportation cost) will be provided to each participant for their safe return to home after a successful completion of providing biological samples. Ethical clearance was obtained from the institutional review board of icddr,b, Research Review Committee (RRC) and Ethical Review Committee (ERC).

## Discussion

These groups of KPs have primarily been resorting to NGO-operated HIV prevention interventions as their first line of healthcare for over two decades in Bangladesh [36]. However, these interventions mainly focus on the prevention (distribution of condoms/lubricants/ attending behavioural change communications, BCC), and treatment and care of HIV and STIs. Although SRHR-related issues are somewhat addressed, these merely exist in the form of referral linkages to mainstream healthcare facilities, where KPs are generally reluctant to visit due to the fear of stigma, discrimination and lack of readiness of healthcare facilities [37]. Moreover, it is also challenging to operate these interventions in a context of negative socio-political and religious sentiments associated with KPs' lifestyles and behaviours [38]. Although the demand to integrate SRHR into HIV programming is mounting, it was not embedded within the HIV prevention modalities at full-scale due to the shortage of funding.

Globally, comprehensive surveillance initiatives on SRHR among these population groups remain scant. Although some systematic reviews, cohort studies, and large-scale surveys have been conducted on KPs, they often focus on specific SRHR issues within a single population group. For instance, there are single cross-sectional studies on maternal health of FSW and specialized surveys on hormone therapy of transgender women [6, 39]. However, these efforts predominantly centre on HIV and STIs, neglecting other of SRHR like sexual health and rights, gender dysphoria, access to SRH services, and stigma and discrimination. Consequently, there is a dearth of data on diverse SRHR issues, particularly among marginalized and hidden groups. The existing SRHR data on KPs are currently fragmented, although it would be beneficial to have a centralised source of SRHR information, so that such services can be tailored accordingly and research can be initiated on emerging SRHR issues, such as 1) availability of services to manage adverse effects of steroids and hormone use by the MSW/hijra, 2) availability of ICT based counselling services for psychosexual and psychosocial complications among MSM/MSW/hijra, 3) services available to provide treatment for etiological STIs among MSM/MSW/hijra/FSW, and 4) availability of counselling on methamphetamine use and relevant SRHR issues among MSM/MSW/hijra/FSW.

In some countries, cohort studies have been conducted among KPs, with some SRHR elements. However, these initiatives have presented their own set of challenges, particularly loss to follow-up. A cohort study was conducted among PWID in Dhaka and Chandpur in Bangladesh. The experience demonstrated that cohort studies were costly, logistically challenging to execute, time-consuming and prone to lost to follow-up. A study has shown that various groups of KPs in Bangladesh are mobile [40, 41] that further fuel loss to follow-up. On the other hand, a surveillance system adopting a repeated cross-sectional method would be a feasible both from logistics and implementation aspects, and can be a robust way of describing the trend of the SRHR situation among KPs.

Yearly data on selective SRHR indicators could serve as an early warning system, underpinning research and intervention priorities. Estimated prevalence data for key SRHR issues, including sexual health, rights, gender dysphoria, access to health services, stigma, and discrimination, at different time points can reveal changes in behavioural trends and emerging concerns among KPs. The surveillance could also measure risk behaviours related to the SRHR health outcomes, which could help redesign and refine KP interventions. While comprehensive SRHR interventions are limited in Bangladesh, some interventions address specific components. The SRHR surveillance could help directly measure the effects of any such SRHR interventions if exist in the study areas, and monitor the changes on selective SRHR indicators if that is possible. Therefore, once established, the SRHR surveillance can provide this opportunity in Bangladesh. Moreover, in the context of the current Covid-19 pandemic, a surveillance initiative could also assess the burden of COVID-19 and its impact on SRHR service uptake by the KPs.

## Dissemination of findings

At the end of each round of data collection, at first, study report will be prepared and shared with the scientists of relevant field in icddr,b. After taking feedback, reports will be improved. Thereafter, a dissemination seminar will be organized to share the results with donor, policy makers and relevant stake holders. In addition to that, findings will also be shared with national and international audience by publishing manuscripts/presentations/etc.

## Study strengths and limitations

The proposed SRHR surveillance has one strength and two limitations. The study will be conducted adopting a mixed method design that will enhance scientific integrity of the study in interpretation of data. Our study has two limitations. 1) The SRHR surveillance will be carried out only in two districts (Dhaka and Jashore) therefore, findings from this study will not represent all MSM/MSW/Hijra/FSW in Bangladesh; 2) time location sampling (TLS) that will be used to collect data from MSM/MSW/FSW will cover only those KPs who are visible in the spots at that time period and hence will ignore those who are hidden/hard to reach.

## Conclusion

This surveillance is the first of its kind to systematically assess the SRHR situation among key populations at risk of HIV, particularly through the blending of quantitative and qualitative methodologies. This study will not only elicit the breadth of the SRHR burden and unmet needs of KPs through quantitative data collection, but will also explore depth and nuanced contexts of the SRHR complexities through qualitative inquiry. The uniqueness of this study not only lies in the methodology but also because it differs from other one-shot, ad hoc research initiatives that have only captured a single-shot scenario of SRHR. It is expected that this study can provide a breadth and depth of insights needed for enriching the SRHR

knowledge base and improving the existing SRHR intervention modalities for these vulnerable and marginalized populations which compromised SRHR outcomes.

## Supporting information

**S1 File. All tables are provided in Annexes 1–5.**
(DOCX)

## Author Contributions

**Conceptualization:** Md. Masud Reza, Golam Sarwar, Samira Dishti Irfan, Mohammad Niaz Morshed Khan, A. K. M. Masud Rana, Muhammad Manwar Morshed Hemel, Mohammad Sha Al Imran, Md. Mahbubur Rahman, Tanveer Khan Ibne Shafiq, Md. Safiullah Sarker, Muntasir Alam, Mustafizur Rahman, Sharful Islam Khan.

**Data curation:** Md. Masud Reza, Samira Dishti Irfan, Mohammad Niaz Morshed Khan, A. K. M. Masud Rana, Muhammad Manwar Morshed Hemel, Mohammad Sha Al Imran, Md. Mahbubur Rahman, Tanveer Khan Ibne Shafiq, Md. Safiullah Sarker, Muntasir Alam, Mustafizur Rahman, Sharful Islam Khan.

**Formal analysis:** Md. Masud Reza, Golam Sarwar, Samira Dishti Irfan, Mohammad Niaz Morshed Khan, A. K. M. Masud Rana, Mohammad Sha Al Imran, Tanveer Khan Ibne Shafiq, Md. Safiullah Sarker, Mustafizur Rahman, Sharful Islam Khan.

**Funding acquisition:** Sharful Islam Khan.

**Investigation:** Md. Masud Reza, Golam Sarwar, Mohammad Niaz Morshed Khan, A. K. M. Masud Rana, Muhammad Manwar Morshed Hemel, Mohammad Sha Al Imran, Muntasir Alam, Sharful Islam Khan.

**Methodology:** Md. Masud Reza, Golam Sarwar, Samira Dishti Irfan, Mohammad Niaz Morshed Khan, A. K. M. Masud Rana, Muhammad Manwar Morshed Hemel, Mohammad Sha Al Imran, Md. Mahbubur Rahman, Tanveer Khan Ibne Shafiq, Md. Safiullah Sarker, Muntasir Alam, Mustafizur Rahman, Sharful Islam Khan.

**Project administration:** Md. Masud Reza, Golam Sarwar, Samira Dishti Irfan, Mohammad Niaz Morshed Khan, A. K. M. Masud Rana, Muhammad Manwar Morshed Hemel, Md. Mahbubur Rahman, Md. Safiullah Sarker, Mustafizur Rahman, Sharful Islam Khan.

**Resources:** Md. Masud Reza, Golam Sarwar, Mohammad Niaz Morshed Khan, Muhammad Manwar Morshed Hemel, Tanveer Khan Ibne Shafiq, Md. Safiullah Sarker, Muntasir Alam, Sharful Islam Khan.

**Software:** Sharful Islam Khan.

**Supervision:** Md. Masud Reza, Golam Sarwar, Samira Dishti Irfan, Mohammad Niaz Morshed Khan, Md. Mahbubur Rahman, Muntasir Alam, Mustafizur Rahman, Sharful Islam Khan.

**Validation:** Md. Masud Reza, Sharful Islam Khan.

**Visualization:** Md. Masud Reza, Sharful Islam Khan.

**Writing – original draft:** Md. Masud Reza, Golam Sarwar, Samira Dishti Irfan, Mohammad Niaz Morshed Khan, A. K. M. Masud Rana, Muhammad Manwar Morshed Hemel, Mohammad Sha Al Imran, Md. Mahbubur Rahman, Tanveer Khan Ibne Shafiq, Md. Safiullah Sarker, Muntasir Alam, Mustafizur Rahman, Sharful Islam Khan.

**Writing – review & editing:** Md. Masud Reza, Golam Sarwar, Samira Dishti Irfan, Moham-
mad Niaz Morshed Khan, A. K. M. Masud Rana, Muhammad Manwar Morshed Hemel,
Mohammad Sha Al Imran, Md. Mahbubur Rahman, Tanveer Khan Ibne Shafiq, Md. Safiul-
lah Sarker, Muntasir Alam, Mustafizur Rahman, Sharful Islam Khan.

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
