## [Decision Letter · Decision Letter 0]

23 May 2023

PONE-D-23-05970Establishing a surveillance system on sexual and reproductive health and rights (SRHR) of key populations (KPs) at risk of compromised outcome of SRHR- A protocol for a mixed-method studyPLOS ONE

Dear Dr. Khan, Thank you for submitting your manuscript to PLOS ONE. After careful consideration, we feel that it has merit but does not fully meet PLOS ONE’s publication criteria as it currently stands. Therefore, we invite you to submit a revised version of the manuscript that addresses the points raised during the review process.

We look forward to receiving your revised manuscript.

Kind regards,

Lebeza Alemu Tenaw

Academic Editor

PLOS ONE

2. You indicated that you had ethical approval for your study. In your Methods section, please ensure you have also stated whether you will obtain consent from parents or guardians of the minors who will be included in the study or whether the research ethics committee or IRB specifically waived the need for their consent.

“The authors greatly thank the Department of Foreign Affairs, Trade and Development (DFATD), Canada for the funding of the study. The authors also thank the Institutional Review Board of icddr,b, for approving the study protocol. icddr,b is also thankful to the Governments of Bangladesh, Canada, Sweden and the UK for providing core/unrestricted support to icddr,b.”

“The funders did not and will not have a role in study design, data collection and analysis, decision to publish, or preparation of the manuscript.”

a) If there are ethical or legal restrictions on sharing a de-identified data set, please explain them in detail (e.g., data contain potentially sensitive information, data are owned by a third-party organization, etc.) and who has imposed them (e.g., an ethics committee). Please also provide contact information for a data access committee, ethics committee, or other institutional body to which data requests may be sent. Please note that authors, including Corresponding Authors, are not permitted to be the sole point of contact for data requests.

b) If there are no restrictions, please provide the minimal anonymized data set necessary to replicate your study findings as either Supporting Information files or to a stable, public repository and provide us with the relevant URLs, DOIs, or accession numbers. For a list of acceptable repositories, please see http://journals.plos.org/plosone/s/data-availability#loc-recommended-repositories.

7. We note that Figure 1  in your submission contain [map/satellite] images which may be copyrighted. All PLOS content is published under the Creative Commons Attribution License (CC BY 4.0), which means that the manuscript, images, and Supporting Information files will be freely available online, and any third party is permitted to access, download, copy, distribute, and use these materials in any way, even commercially, with proper attribution. For these reasons, we cannot publish previously copyrighted maps or satellite images created using proprietary data, such as Google software (Google Maps, Street View, and Earth). For more information, see our copyright guidelines: http://journals.plos.org/plosone/s/licenses-and-copyright.

Additional Editor Comments:

Abstract:

- Minimize abbreviations in the abstract

Introduction and Methods,

- There are bulky paragraphs which needs paraphrasing like Line 112-124, 128-140, 263-276, 332-344, 376-386, 412-423, 443-454, 508-525

- The statements " Moreover, SRHR-related research has mainly focused on women, despite the importance of male sexual health. However, it is not possible to improve their overall health and quality of life if the issues related to SRHR of male and female KPs are not addressed properly and simultaneously' needs reference.

- There is no clear description about the study design and data collection techniques for each specific objective.

- Why did you used Finite Population Correction (FPC) for adjustment of sample size?

- Minimize statistical formulas in the sample size calculation

Reviewers' comments:

Reviewer's Responses to Questions

**Comments to the Author**

1. Does the manuscript provide a valid rationale for the proposed study, with clearly identified and justified research questions?

Reviewer #1: Yes

2. Is the protocol technically sound and planned in a manner that will lead to a meaningful outcome and allow testing the stated hypotheses?

Reviewer #1: Yes

3. Is the methodology feasible and described in sufficient detail to allow the work to be replicable?

Reviewer #1: Yes

4. Have the authors described where all data underlying the findings will be made available when the study is complete?

Reviewer #1: No

5. Is the manuscript presented in an intelligible fashion and written in standard English?

Reviewer #1: No

6. Review Comments to the Author

You may also provide optional suggestions and comments to authors that they might find helpful in planning their study.

Reviewer #1: Abbreviations should not be used in the abstract.

Abstract: Line 28: ‘’SRHR’’ use the full form upon initial usage.

Background: Line 77-78: “UN’’, ‘’UNFPA”, and “UNHCR’’ at first, use a full form.

Background: Line 88, 215: It is not desirable to begin a paragraph with an abbreviation

Background: Line 90: “UNHRC’’ At first, use a full form

In the background section, there are long sentences from a single reference. Try to conduct a thorough review and integrate your paragraphs from different literature.

Your primary objective and the first secondary objective are the same. Why?

In the ethics statement: ‘’ The verbal consents of the participants were obtained’’. This statement is incorrect and should be corrected.

Line 189, ‘’ Assent will be taken from those who will be 15 to less than 18 years of age’’ Are you going to take assent from the participant itself?

Avoid long paragraphs throughout the manuscript.

Be cautious about the usage of abbreviations throughout the manuscript.

7. PLOS authors have the option to publish the peer review history of their article (what does this mean?). If published, this will include your full peer review and any attached files.

Reviewer #1: No

---

## [Author Response · Author response to Decision Letter 0]

6 Jul 2023

Response to the Reviewers’ Comments

Response: In the first submission, by mistake, the font style of the manuscript was in Calibri with font size 11. However, this time we have made it into Times New Roman with 12 font size and therefore, the line number increased. 

2. You indicated that you had ethical approval for your study. In your Methods section, please ensure you have also stated whether you will obtain consent from parents or guardians of the minors who will be included in the study or whether the research ethics committee or IRB specifically waived the need for their consent.

Response: Informed written assent will be taken from the participants of 15 to less than 18 years of age and informed written consent will be taken from those who will be 18 years or above years of age. Written assent will be taken from local gatekeepers (such as, hawkers in the spots or someone who belongs to the community of the participant and at the same time knows the participant very well and older in age, at least 18 years, than the participant) whose help will be sought to identify participants. The assent will be read out to the participant and gatekeeper and their signature or left thumb print impression will be taken on the survey form. 

This is to be mentioned that in the context of Bangladesh, the target population groups are highly stigmatised and their sexual behaviour (male-to-male sex) is not only culturally unaccepted but also prohibited by law so that there is no way we can identify and conduct face-to-face interview at their house where they live with their parents/siblings/other family members. Therefore, based on the context of Bangladesh, the process of taking consent and assent that is being followed in our study is being practised in conducing HIV surveillance rounds among KPs in Bangladesh over the years in the past.

Response: A separate funding statement is prepared to be submitted. Also, we have included funding information in the cover letter. 

“The authors greatly thank the Department of Foreign Affairs, Trade and Development (DFATD), Canada for the funding of the study. The authors also thank the Institutional Review Board of icddr,b, for approving the study protocol. Icddr,b is also thankful to the Governments of Bangladesh, Canada, Sweden and the UK for providing core/unrestricted support to icddr,b.”

“The funders did not and will not have a role in study design, data collection and analysis, decision to publish, or preparation of the manuscript.”

Response: We have taken out the ‘Acknowledgement section” from the manuscript and provided that in the cover letter. 

a) If there are ethical or legal restrictions on sharing a de-identified data set, please explain them in detail (e.g., data contain potentially sensitive information, data are owned by a third-party organization, etc.) and who has imposed them (e.g., an ethics committee). Please also provide contact information for a data access committee, ethics committee, or other institutional body to which data requests may be sent. Please note that authors, including Corresponding Authors, are not permitted to be the sole point of contact for data requests.

b) If there are no restrictions, please provide the minimal anonymized data set necessary to replicate your study findings as either Supporting Information files or to a stable, public repository and provide us with the relevant URLs, DOIs, or accession numbers. For a list of acceptable repositories, please see http://journals.plos.org/plosone/s/data-availability#loc-recommended-repositories.

Response: We did not use any data to prepare our manuscript therefore, these restrictions do not apply to our manuscript. However, the objectives of the proposed SRHR surveillance are to understand the burden, unmet needs and the overall situation of SRHR among MSM, MSW, transgender women (locally known as hijra) and FSW in selected areas in Bangladesh. This type of surveillance study, adopting a mixed-method, is being conducted for the first time not only in Bangladesh but also globally. We will collect data twice in year-1 (2022) and year-2 (2023). In the budget, we kept some money for publications where primary data from this surveillance study will be used to prepare manuscripts. We know that PLOS ONE publishes protocols that reach a variety of readers. Therefore, we thought that we also take an initiative to publish the protocol that may have huge impact in improving the wellbeing and the future design of SRHR services among the targeted population groups in other countries. This is to be noted that as soon as data are ready, we will start work to prepare manuscripts on various SRHR issues of the studied population groups for publications and we hope that we will submit manuscripts to the PLOS ONE. 

Response: Our study was approved by the Research Review Committee (RRC) and Ethical Review Committee (ERC) of icddr,b. Mr. Shafiqul Alam Sarker, MD, Ph.D, FRCP is the Chairperson of RRC and Professor Ahmed Abu Saleh is the Chairperson of ERC. During submission, we had attached the approval copies from RRC and ERC. 

7. We note that Figure 1 in your submission contain [map/satellite] images which may be copyrighted. All PLOS content is published under the Creative Commons Attribution License (CC BY 4.0), which means that the manuscript, images, and Supporting Information files will be freely available online, and any third party is permitted to access, download, copy, distribute, and use these materials in any way, even commercially, with proper attribution. For these reasons, we cannot publish previously copyrighted maps or satellite images created using proprietary data, such as Google software (Google Maps, Street View, and Earth). For more information, see our copyright guidelines: http://journals.plos.org/plosone/s/licenses-and-copyright.

Response: We have removed Figure-1 from the manuscript. 

Additional Editor Comments:

Abstract:

- Minimize abbreviations in the abstract

Response: All abbreviations have been removed in the abstract. 

Introduction and Methods,

- There are bulky paragraphs which needs paraphrasing like Line 112-124, 128-140, 263-276, 332-344, 376-386, 412-423, 443-454, 508-525

Response: We have taken care of these by rewriting the lines. 

- The statements " Moreover, SRHR-related research has mainly focused on women, despite the importance of male sexual health. However, it is not possible to improve their overall health and quality of life if the issues related to SRHR of male and female KPs are not addressed properly and simultaneously' needs reference.

Response: The reference has been added in the manuscript. 

- There is no clear description about the study design and data collection techniques for each specific objective.

Response: We have added Annex 5 to explain this. Please look at the file of All Annexes. 

- Why did you used Finite Population Correction (FPC) for adjustment of sample size?

Response: Since the survey will be conducted without replacement from a finite population therefore, we followed using FPC in the sample size calculation as suggested by Leslie Kish (reference#23 in the manuscript). 

- Minimize statistical formulas in the sample size calculation. 

Response: Statistical formulas have been minimized in the manuscript. 

Reviewers' comments:

Reviewer's Responses to Questions

Comments to the Author

1. Does the manuscript provide a valid rationale for the proposed study, with clearly identified and justified research questions?

Reviewer #1: Yes

 2. Is the protocol technically sound and planned in a manner that will lead to a meaningful outcome and allow testing the stated hypotheses?

 Reviewer #1: Yes

 3. Is the methodology feasible and described in sufficient detail to allow the work to be replicable?

 Reviewer #1: Yes

 4. Have the authors described where all data underlying the findings will be made available when the study is complete?

 Reviewer #1: No

 5. Is the manuscript presented in an intelligible fashion and written in standard English?

 Reviewer #1: No

 6. Review Comments to the Author

You may also provide optional suggestions and comments to authors that they might find helpful in planning their study.

Reviewer #1: Abbreviations should not be used in the abstract.

Response: All abbreviations have been removed in the abstract. 

Abstract: Line 28: ‘’SRHR’’ use the full form upon initial usage.

Response: Full form of SRHR is provided at the first usage. 

Background: Line 77-78: “UN’’, ‘’UNFPA”, and “UNHCR’’ at first, use a full form.

Response: Full form of “UN’’, ‘’UNFPA”, and “UNHCR’’ is provided. 

Background: Line 88, 215: It is not desirable to begin a paragraph with an abbreviation

Response: The line has been edited. 

Background: Line 90: “UNHRC’’ At first, use a full form

Response: Full form of “UNHRC’’ is provided. 

In the background section, there are long sentences from a single reference. Try to conduct a thorough review and integrate your paragraphs from different literature.

Response: This has been taken care of as you have suggested. 

Your primary objective and the first secondary objective are the same. Why?

Response: The objectives have been edited. Thank you for the notification. 

In the ethics statement: ‘’ The verbal consents of the participants were obtained’’. This statement is incorrect and should be corrected.

Response: The line has been edited. 

Line 189, ‘’ Assent will be taken from those who will be 15 to less than 18 years of age’’ Are you going to take assent from the participant itself?

Response: Informed written assent will be taken from the participants of 15 to less than 18 years of age and informed written consent will be taken from those who will be 18 years or above years of age. Written assent will be taken from local gatekeepers (such as, hawkers in the spots or someone who belongs to the community of the participant and at the same time knows the participant very well and older in age, at least 18 years, than the participant) whose help will be sought to identify participants. The assent will be read out to the participant and gatekeeper and their signature or left thumb print impression will be taken on the survey form. 

This is to be mentioned that in the context of Bangladesh, the target population groups are highly stigmatised and their sexual behaviour (male-to-male sex) is not only culturally unaccepted but also prohibited by law so that there is no way we can identify and conduct face-to-face interview at their house where they live with their parents/siblings/other family members. Therefore, based on the context of Bangladesh, the process of taking consent and assent that is being followed in our study is being practised in conducing HIV surveillance rounds among KPs in Bangladesh over the years in the past.

Avoid long paragraphs throughout the manuscript.

Response: This has been taken care of as you have suggested. 

Be cautious about the usage of abbreviations throughout the manuscript.

Response: All abbreviations are checked and actions taken as suggested. 

 7. PLOS authors have the option to publish the peer review history of their article (what does this mean?). If published, this will include your full peer review and any attached files.

Response: We prefer to choose ‘yes’.

Do you want your identity to be public for this peer review? For information about this choice, including consent withdrawal, please see our Privacy Policy.

 Reviewer #1: No________________________________________While revising your submission, please upload your figure files to the Preflight Analysis and Conversion Engine (PACE) digital diagnostic tool, https://pacev2.apexcovantage.com/. PACE helps ensure that figures meet PLOS requirements. To use PACE, you must first register as a user. Registration is free. Then, login and navigate to the UPLOAD tab, where you will find detailed instructions on how to use the tool. If you encounter any issues or have any questions when using PACE, please email PLOS at figures@plos.org. Please note that Supporting Information files do not need this step.

---

## [Editor Report · Decision Letter 1]

10 Jul 2023

Establishing a surveillance system on sexual and reproductive health and rights (SRHR) of key populations (KPs) at risk of compromised outcome of SRHR- A protocol for a mixed-method study

PONE-D-23-05970R1

Dear Dr. Sharful,

We’re pleased to inform you that your manuscript has been judged scientifically suitable for publication and will be formally accepted for publication once it meets all outstanding technical requirements.

Kind regards,

Lebeza Alemu Tenaw

Academic Editor

PLOS ONE

---

## [Editor Report · Acceptance letter]

19 Jul 2023

PONE-D-23-05970R1 

Establishing a surveillance system on sexual and reproductive health and rights (SRHR) of key populations (KPs) at risk of compromised outcome of SRHR- A protocol for a mixed-method study 

Dear Dr. Khan:

I'm pleased to inform you that your manuscript has been deemed suitable for publication in PLOS ONE. Congratulations! Your manuscript is now with our production department. 

Kind regards, 

on behalf of

Mr. Lebeza Alemu Tenaw 

Academic Editor

PLOS ONE